# Predictors for self-discontinuation of anti-osteoporosis medication: A hospital-based real-world study

Ya-Lian Deng[1☯], Chun-Sheng Hsu[2,3,4☯], Chiann-Yi Hsu[5], Chih-Hui Chen[6,7], Shiang-Ferng Ou[1], Chin-Feng Liu[1], Shu-Hui Yang[1], Chun-Hsi Shih[8], Yi-Ming Chen[3,5,7,9,10‡]*, Hsu-Tung Lee[3,11,12,13‡]

1 Department of Nursing, Taichung Veterans General Hospital, Taichung, Taiwan, 2 Department of Physical Medicine and Rehabilitation, Taichung Veterans General Hospital, Taichung, Taiwan, 3 Department of Post-Baccalaureate Medicine, College of Medicine, National Chung Hsing University, Taichung, Taiwan, 4 School of Medicine, National Defense Medical Center, Taipei, Taiwan, 5 Department of Medical Research, Taichung Veterans General Hospital, Taichung, Taiwan, 6 Department of Orthopedics, Taichung Veterans General Hospital, Taichung, Taiwan, 7 School of Medicine, College of Medicine, National Yang Ming Chiao Tung University, Taipei, Taiwan, 8 Department of Radiology, Taichung Veterans General Hospital, Taichung, Taiwan, 9 Division of Allergy, Immunology and Rheumatology, Taichung Veterans General Hospital, Taichung, Taiwan, 10 Rong Hsing Research Center for Translational Medicine & Ph.D. Program in Translational Medicine, National Chung Hsing University, Taichung, Taiwan, 11 Cancer Prevention and Control Center, Taichung Veterans General Hospital, Taichung, Taiwan, 12 Department of Neurosurgery, Neurological Institute, Taichung Veterans General Hospital, Taichung, Taiwan, 13 Graduate Institute of Medical Sciences, National Defense Medical Center, Taipei, Taiwan

☯ These authors contributed equally to this work.
‡ YMC and HTL also contributed equally to this work.
* ymchen1@vghtc.gov.tw

**Data Availability Statement:** Data are available from the Ethics Committee of Taichung Veterans General Hospital for researchers who meet the

## Abstract

Osteoporotic fractures have a tremendous impact on quality of life and may contribute to fatality, but half of patients may discontinue their anti-osteoporosis medication. The study aimed to investigate the factors associated with the persistence of anti-osteoporosis medication. Between June 2016 and June 2018, we recruited 1195 participants discontinuing prior anti-osteoporosis medication. Telephone interviews were conducted to discern the reasons for discontinuation. Comparisons among groups and risks of self-discontinuation were analyzed. Among 694 patients who have no records of continuing anti-osteoporosis medication, 374 (54%) self-discontinued, 64 (9.2%) discontinued due to physicians' suggestion, and 256 (36.8%) with unintended discontinuation. Among patients with self-discontinuation, 173 (46.3%) forgot to visit outpatient clinics; 92 (24.5%) discontinued because of medication-related factors; 57 (15.2%) thought the severity of osteoporosis had improved and therefore discontinued; 30 (8%) stopped due to economic burden; 22 (5.9%) were lost to follow-up because of newly diagnosed diseases other than osteoporosis. Additionally, older age, male gender, calcium supplement, teriparatide therapy and hip fractures in teriparatide users were associated with adherence to anti-osteoporosis drugs. In conclusion, our results indicate that younger age, female gender, non-use of calcium supplements, and anti-resorptive medication were independent risk factors associated with drug discontinuation. Identifying high-risk patients and providing timely health education are crucial for adherence to anti-osteoporosis medication.

criteria for access to these data by contacting irbtc@mail.vghtc.gov.tw.

**Funding:** The authors received no specific funding for this work.

**Competing interests:** The authors have declared that no competing interests exist.

## 1. Introduction

Osteoporosis is a chronic disease characterized by low bone mass and deterioration of the microstructure of bone tissues, leading to increased bone fragility and a subsequent increased risk of fractures [1, 2]. Osteoporotic fractures are currently a major problem for the elderly population, especially elderly women. As the size of geriatric populations increases, the impact of osteoporosis is expected to grow. In addition to increased morbidity, mortality, and poor health-related quality of life, fragility fractures also place a substantial financial burden on the health care system [3–7]. Therefore, it is imperative to screen and manage osteoporosis to prevent comorbid conditions associated with fragility fracture.

The essential goal of osteoporosis treatment is to prevent osteoporotic fractures. Poor adherence to osteoporosis drugs is associated with reduced efficacy of anti-osteoporosis medication, leading to fractures, increased medical expenditures, and mortality [5–9]. Moreover, a prior report also showed that improved osteoporosis medication adherence can reduce osteoporosis-related health care costs by preventing fractures [8]. A study conducted in France demonstrated that patients with higher adherence to osteoporosis medication exhibited a 28–32% reduction in fracture risk compared with patients with poor adherence [10]. However, patients with osteoporosis may discontinue anti-osteoporosis medication for various reasons. In a study of 191 patients who discontinued oral bisphosphonates, the leading cause of drug discontinuation was adverse events (43.9%) [11].

In addition, worrying about the occurrence of medication-related adverse events, not feeling the effectiveness of the treatment, and the cost of the drug were also associated with the persistence of anti-osteoporosis medication [11]. In patients with rheumatoid arthritis poor compliance with oral bisphosphonates could limit the therapeutic efficacy [12]. Extended dosing frequency of anti-osteoporosis medication was associated with superior persistence, with monthly use of bisphosphonates better than weekly regimens [12]. In addition, patients initiating an every-6-month injection had significantly higher persistence compared with those initiating more frequently dosed oral or injectable agents [13, 14]. Moreover, gastrointestinal adverse events and poor health literacy were the main causes of medication discontinuation. Therefore, proactive patient education could help improve adherence to osteoporosis medication [12]. Patients enrolled in dedicated health education and follow-up programs after initiating anti-osteoporosis drugs improved medication persistence [15, 16]. However, most previous studies of osteoporosis medication adherence are based on oral bisphosphonates. Investigations of osteoporosis medication involving other mechanisms of action are scanty. Furthermore, predictive factors for discontinuation of osteoporosis medication have not been explored.

The primary aim was to investigate the causes of anti-osteoporosis medication discontinuation. The secondary aim was to explore risk factors for self-discontinuation among patients who stopped taking anti-osteoporosis medication.

## 2. Materials and methods

### 2.1 Enrolled participants

We conducted a retrospective data analysis using the osteoporosis database of Taichung Veterans General Hospital, Taiwan. Our osteoporosis database enrolled osteoporotic patients identified through the presence of the International Classification of Disease, 9th Revision, Clinical Modification (ICD-9-CM) and 10th edition (ICD-10) diagnostic code for osteoporosis (733.0, 733.00, 733.01, 733.02, 733.03, 733.09/M80–82) or for osteoporotic fractures, including vertebral fractures (805.2–805.9/S22, S32), hip fractures (820.x/S72), humeral (812.x/S42), and

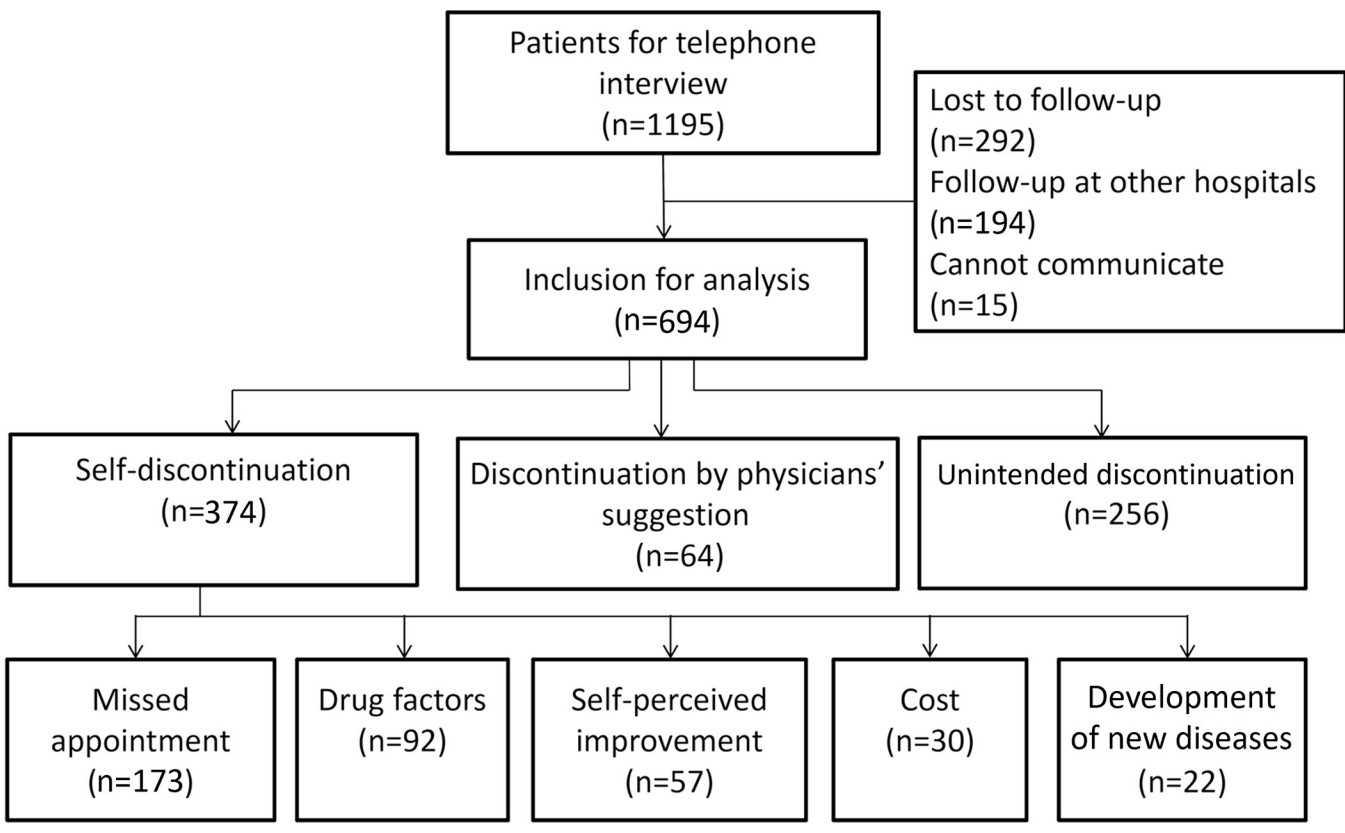

**Fig 1. Enrolled participants flow charts.**

radio-ulnar fractures (813.x/S52, S62) on medical records. Patients with a history of traumatic accidents were excluded. In this study, a total of 1195 patients with prior exposure to anti-osteoporosis medication that was discontinued between June 2016 and June 2018 were enrolled. After conducting consistency training of the nursing staff, patients who have no records of continuing anti-osteoporosis medication were interviewed by telephone to determine the reasons for discontinuation, to provide health education of osteoporosis management, and to assist with outpatient clinic visit appointments. The exclusion criteria were patients who could not be contacted, patients who had taken anti-osteoporosis medication in other hospitals, and patients who were unable to communicate (n = 501, Fig 1). In this study, denosumab discontinuation was defined as the absence of denosumab claim more than 6 months + 8 weeks since the last dose and without any anti-osteoporosis medication other than denosumab within 6 months + 8 weeks after prior denosumab treatment [17, 18]. Teriparatide discontinuation was defined as the absence of any anti-osteoporosis medication more than 8 weeks after a prior dose of teriparatide and the duration of teriparatide treatment after the first dose was limited to less than 18 months [19, 20]. Bisphosphonates or raloxifene discontinuation were defined as ≥12 months without any anti-osteoporosis medication prescription claims after prior bisphosphonate or raloxifene treatment [21]. Demographic data, sites of fractures, BMD, prior anti-osteoporosis medication before discontinuation, and comorbidities were then traced retrospectively from the electronic health records. This study was approved by the Ethics Committee of Clinical Research, Taichung Veterans General Hospital (CF18067B). As the patients' data were anonymized before analysis, the requirement for written consent from patients was waived for this study and approved by the Ethics Committee.

## 2.2 Causes of discontinuation of anti-osteoporosis medication

The reasons for discontinuation events are divided into three categories. First, self-discontinuation by the patient; second, discontinuation of the anti-osteoporosis medication due to the physician's recommendation including bisphosphonates holidays; third, unintended discontinuation due to mortality if the participants had good adherence to anti-osteoporotic medication before death. In addition, the reasons for self-discontinuation included forgetting to come back for an outpatient visit, medication-related factors, the disease seemed to have improved as judged by the patient, economic burden, and other diseases had developed.

## 2.3 Anti-osteoporotic treatment

The medication records of the enrolled patients from June 2016 until June 2018 were reviewed to identify those who had ever received anti-osteoporotic treatment and the type of medication they received. The anti-osteoporotic medications identified were bisphosphonates (alendronate, and zoledronic acid), selective estrogen receptor modulator (raloxifene), recombinant human parathyroid hormone (teriparatide), and receptor activator of nuclear factor κ-B ligand (RANKL) inhibitor (denosumab). For those with multiple anti-osteoporotic medications, the last type of drug before discontinuation of anti-osteoporosis medication was recorded.

## 2.4 Fracture identification

The data on osteoporotic fractures occurrence and their fracture sites, for fractures sustained between June 2016 and June 2018, were extracted from Taichung Veterans General Hospital's medical records and radiographic reports. Osteoporotic fractures included vertebrae spine fractures, hip fractures, distal radius fractures such as Colles' fractures, and fractures of other bones, such as the proximal humerus, distal radius, ribs, tibia-fibula, patella, and pelvis. Participants with fractures due to vehicular accidents or high-impact trauma (ICD-9-CM code E810–E819, E881–E883, E884), pathological fractures (733.14, 733.15/M84), and those with a diagnosis of Paget's disease (731.0/M88) were excluded for further analysis.

## 2.5 Bone mineral density (BMD) measurements

BMD measurements of the bilateral femoral necks and lumbar spine (L1-L4) were obtained through dual-energy X-ray absorptiometry (DXA), using the Lunar Prodigy (General Electric, Fairfield, CT, USA), with the results expressed in $g/cm^2$. BMD measurement was performed before using anti-osteoporotic medications. The least significant change was $\pm 0.010 g/cm^2$ for the lumbar spine (L1-L4), and $\pm 0.012 g/cm^2$ for the femoral neck. T-scores were determined according to the manufacturer's reference data. According to the WHO criteria, osteoporosis is defined as a T-score $\leqq$ -2.5; low bone mass is defined as a T-score between -1.0 and -2.5; and normal is defined as a T-score > -1.0 [22].

## 2.6 Statistical analysis

The demographic data of the continuous parameters are shown as median (interquartile range, IQR), and for the categorical variables as the number (percent) of patients. A Kruskal-Wallis test was used to compare variables amongst patients with various types of anti-osteoporotic medication and causes of drug discontinuation. Post-hoc analyses were calculated by the Dunn-Bonferroni test. Logistic regression analysis was used to investigate independent factors associated with self-discontinuation of anti-osteoporotic medication and stratified by teriparatide and anti-resorptive therapies. All data were analyzed using the Statistical Package for the Social Sciences (SPSS) version 22.0. Significance was set at $p < 0.05$.

## 3. Results

### 3.1 Patient characteristics by discontinued anti-osteoporosis medication

A total of 694 patients who discontinued osteoporosis medication were enrolled in this study (Table 1). One hundred and sixty-five patients discontinued Raloxifene, 175 discontinued Alendronate, 38 discontinued Zoledronic acid, 286 discontinued Denosumab, and 30 discontinued Teriparatide. Among those who discontinued Teriparatide, their body height was

**Table 1. Patient characteristics by discontinued anti-osteoporosis medication.**

| | Raloxifene (n = 165) | | Alendronate (n = 175) | | Zoledronic acid (n = 38) | | Denosumab (n = 286) | | Teriparatide (n = 30) | | p value |
|---|---|---|---|---|---|---|---|---|---|---|---|
| Age | 75.6 (68.0–84.1) | | 77.9 (68.3–87.0) | | 77.6 (71.4–87.4) | | 79.0 (72.3–85.2) | | 78.1 (74.3–85.2) | | 0.152 |
| Female | 165 (100%) | | 121 (69.1%) | | 25 (65.8%) | | 217 (75.9%) | | 29 (96.7%) | | <0.001 3[#¤ ¢ © £ *] |
| Body mass index (kg/m$^2$) | 23.1 (20.2–26.0) | | 22.7 (20.6–25.7) | | 23.0 (20.2–25.8) | | 23.3 (20.6–26.2) | | 23.0 (20.7–24.6) | | 0.846 |
| Smoking | 6 (3.7%) | | 27 (15.5%) | | 5 (13.2%) | | 32 (11.4%) | | 2 (6.7%) | | 0.008 [#¢] |
| Alcohol consumption | 2 (1.9%) | | 14 (12.4%) | | 1 (4.4%) | | 17 (8.1%) | | 2 (8.7%) | | 0.066 |
| Fracture sites | | | | | | | | | | | |
| Hip | 15 (9.1%) | | 34 | (19.4%) | 5 | (13.2%) | 60 | (21.0%) | 9 | (30.0%) | 0.006 [#¢ §] |
| Spine | 113 (68.5%) | | 115 | (65.7%) | 23 | (60.5%) | 198 | (69.2%) | 26 | (86.7%) | 0.171 |
| Radius | 8 (4.9%) | | 5 | (2.9%) | 0 | (0%) | 7 | (2.5%) | 1 | (3.3%) | 0.495 |
| Humerus | 0 (0%) | | 4 | (2.3%) | 0 | (0%) | 10 | (3.5%) | 1 | (3.3%) | 0.131 |
| Other | 15 (9.1%) | | 35 | (20.0%) | 5 | (13.2%) | 36 | (12.6%) | 3 | (10.0%) | 0.049 |
| Bone mineral density(g/cm$^2$) | | | | | | | | | | | |
| Lumbar spine | 0.800 | (0.7–0.9) | 0.846 | (0.7–1) | 0.788 | (0.6–0.9) | 0.782 | (0.6–0.9) | 0.696 | (0–0.8) | 0.011 [©] |
| Right femoral neck | 0.618 | (0.5–0.7) | 0.617 | (0.5–0.7) | 0.622 | (0.5–0.7) | 0.595 | (0.5–0.7) | 0.525 | (0.3–0.6) | 0.020 [§ ©] |
| Left femoral neck | 0.625 | (0.5–0.7) | 0.624 | (0.5–0.7) | 0.640 | (0.6–0.7) | 0.601 | (0.5–0.7) | 0.498 | (0–0.6) | <0.001 [§ © £ *] |
| Calcium supplement | 101 | (61.2%) | 108 | (61.7%) | 24 | (63.2%) | 202 | (70.6%) | 15 | (50.0%) | 0.065 |
| Vitamin D supplement | 28 | (17.0%) | 19 | (10.9%) | 5 | (13.2%) | 91 | (31.8%) | 5 | (16.7%) | <0.001 [¢ ¥] |
| Comorbidity | | | | | | | | | | | |
| Osteoarthritis | 114 | (69.1%) | 117 | (66.9%) | 28 | (73.7%) | 195 | (68.2%) | 21 | (70.0%) | 0.943 |
| Rheumatoid arthritis | 11 | (6.7%) | 15 | (8.6%) | 4 | (10.5%) | 30 | (10.5%) | 1 | (3.3%) | 0.521 |
| Diabetes mellitus | 39 | (23.6%) | 47 | (26.9%) | 11 | (29.0%) | 91 | (31.8%) | 9 | (30.0%) | 0.443 |
| Hypertension | 83 | (50.3%) | 100 | (57.1%) | 23 | (60.5%) | 177 | (61.9%) | 15 | (50.0%) | 0.160 |
| Stroke | 26 | (15.8%) | 34 | (19.4%) | 7 | (18.4%) | 63 | (22.0%) | 6 | (20.0%) | 0.617 |
| Hyperthyroidism | 13 | (7.9%) | 12 | (6.9%) | 2 | (5.3%) | 22 | (7.7%) | 2 | (6.7%) | 0.979 |
| Chronic liver disease | 25 | (15.2%) | 42 | (24.0%) | 7 | (18.4%) | 61 | (21.3%) | 6 | (20.0%) | 0.348 |

By Kruskal Wallis test.

Post-hoc analysis by Dunn-Bonferroni test.

p value < 0.05

[#]Raloxifene vs. Alendronate

[¤]Raloxifene vs. Zoledronic acid

[¢]Raloxifene vs. Denosumab

[§]Raloxifene vs. Teriparatide

[µ]Alendronate vs. Zoledronic acid

[¥]Alendronate vs. Denosumab

[©]Alendronate vs. Teriparatide

[à]Zoledronic acid vs. Denosumab

[£]Zoledronic acid vs. Teriparatide

[*]Denosumab vs. Teriparatide

shorter, they had more hip fractures, lower BMD of the lumbar spine, and bilateral femoral necks compared with their counterparts. No significant differences in age, weight, body mass index, alcohol consumption, and comorbidities could be observed among the groups.

## 3.2 Comparisons of demographics among participants discontinued anti-osteoporosis medication due to different causes

In our study, 374 patients (54%) discontinued anti-osteoporosis medication for their own reasons; 64 (9.2%) stopped because of the physicians' recommendation; 256 (36.8%) discontinued due to unintended discontinuation (Table 2). Meanwhile, the reasons for self-discontinuation included 173 (46.3%) who forgot to return to the outpatient clinics; 92 (24.5%) discontinued because of drug-related factors; 57 (15.2%) thought that their osteoporosis had improved; 30 (8%) discontinued due to financial burden; 22 (5.9%) discontinued because they had developed other diseases. We found that female gender, fracture sites other than the hip, and use of raloxifene were correlated with a higher likelihood of self-discontinuation of anti-osteoporosis medication compared with the other groups.

## 3.3 Predictors for self-discontinuation of anti-osteoporosis medication

To determine the risk factors for self-discontinuation of anti-osteoporosis medication, logistic regression analysis was performed (Table 3). We found that older age (OR: 0.95, 95% CI: 0.93–0.96, $p < 0.001$), male gender (OR: 0.48, 95% CI: 0.31–0.74, $p < 0.001$), calcium supplement (OR: 0.67, 95% CI: 0.48–0.95, $p = 0.023$) and use of Teriparatide (OR: 0.19, 95% CI: 0.08–0.49, $p < 0.001$) were associated with lower risks of self-discontinuation.

## 3.4 Predictors for self-discontinuation of anti-resorptive therapy and teriparatide

Fig 2 shows the risk factors of self-discontinuation of anti-osteoporosis medication by mechanisms of action. Older age (OR: 0.95, 95% CI: 0.94–0.97, $p < 0.01$), male gender (OR: 0.42, 95% CI: 0.27–0.65, $p < 0.01$), and calcium supplement (OR: 0.68, 95% CI: 0.47–0.998, $p < 0.05$) were associated with a lower likelihood of discontinuing anti-resorptive therapy. Patients with hip fractures (OR: 0.26, 95% C.I.: 0.07–0.97, $p < 0.05$) were less likely to stop using Teriparatide.

## 3.5 Intervention outcome

After telephone interviews and health education, 22% of the participants resumed anti-osteoporosis medication and visited the outpatient clinics again for follow-up.

## 4. Discussions

In this study, we found that younger age, female, non-use of calcium supplements, and prescription of anti-resorptive therapy were associated with poor adherence to osteoporosis medication. Our results suggest that case management of osteoporosis is necessary to avoid loss to follow-up and discontinuation of essential treatment. Physicians should identify individuals at high risk of discontinuing medication in order to ensure drug persistence and prevent subsequent fractures.

It has been reported that patients with osteoporosis may stop oral bisphosphonates due to adverse events (47.5%), poor health literacy (40.5%), and cost (12%) [12]. Gonnelli et al. also demonstrated that drug-related adverse reactions (43.8%), fear of side effects (23.3), and absence of perception of effectiveness (15.8%) contributed to the major causes for discontinuation of oral bisphosphonates [11]. Previous studies mostly explored compliance with oral

**Table 2. Comparisons of demographics among participants discontinued anti-osteoporosis medication due to different causes.**

| | Self-discontinuation (n = 374) | | Discontinuation by physicians' suggestion (n = 64) | | Unintended discontinuation (n = 256) | | *p* value | *p* value[§] | | |
| --- | --- | --- | --- | --- | --- | --- | --- | --- | --- | --- |
| | | | | | | | | S vs D | S vs U | D vs U |
| Age | 75.0 | (66.7–81.3) | 73.2 | (64.9–78.5) | 84.0 | (77.4–89.3) | <0.001** | 0.673 | <0.001** | <0.001** |
| Gender Female | 324 | (86.6%) | 54 | (84.4%) | 179 | (69.9%) | <0.001** | 0.696 | <0.001** | 0.029* |
| Body mass index (kg/m$^2$) | 23.3 | (20.7–26.5) | 23.7 | (21–25.2) | 22.2 | (19.9–25.1) | 0.006** | 1.000 | 0.004** | 0.409 |
| Smoking | 30 | (8.1%) | 6 | (9.5%) | 36 | (14.3%) | 0.042* | 0.626 | 0.050 | 0.614 |
| Alcohol consumption | 18 | (7.5%) | 2 | (4.3%) | 16 | (8.7%) | 0.595 | | | |
| Fracture sites | | | | | | | | | | |
| Hip | 45 | (12.0%) | 12 | (18.8%) | 66 | (25.8%) | <0.001** | <0.001** | <0.001** | 0.260 |
| Spine | 244 | (65.2%) | 36 | (56.3%) | 195 | (76.2%) | 0.001** | 0.205 | 0.005** | 0.005** |
| Radius | 10 | (2.7%) | 2 | (3.1%) | 9 | (3.5%) | 0.834 | | | |
| Humerus | 7 | (1.9%) | 5 | (7.8%) | 3 | (1.2%) | 0.004** | 0.030* | 0.747 | 0.029* |
| Other | 46 | (12.3%) | 12 | (18.8%) | 36 | (14.1%) | 0.367 | | | |
| Bone mineral density(g/cm$^2$) | | | | | | | | | | |
| Lumbar spine | 0.815 | (0.7–0.9) | 0.836 | (0.7–1.0) | 0.775 | (0.5–0.9) | 0.012* | 0.500 | 0.027* | 0.081 |
| Right femoral neck | 0.628 | (0.6–0.7) | 0.612 | (0.5–0.7) | 0.553 | (0–0.6) | <0.001** | 0.165 | <0.001** | 0.078* |
| Left femoral neck | 0.635 | (0.6–0.7) | 0.612 | (0.5–0.7) | 0.558 | (0–0.6) | <0.001** | 1.000 | <0.001** | 0.012* |
| Calcium supplement | 227 | (60.7%) | 39 | (60.9%) | 184 | (71.9%) | 0.011* | 1.000 | 0.011* | 0.144* |
| Vitamin D supplement | 73 | (19.5%) | 13 | (20.3%) | 62 | (24.2%) | 0.352 | | | |
| Medications | | | | | | | <0.001** | <0.001** | 0.008** | <0.001** |
| Alendronate | 92 | (24.6%) | 28 | (43.8%) | 55 | (21.5%) | 0.004** | | | |
| Denosumab | 149 | (39.8%) | 17 | (26.6%) | 120 | (46.9%) | 0.066 | | | |
| Zoledronic acid | 21 | (5.6%) | 5 | (7.8%) | 12 | (4.7%) | 0.732 | | | |
| Raloxifene | 105 | (28.1%) | 7 | (10.9%) | 53 | (20.7%) | 0.020* | | | |
| Teriparatide | 7 | (1.9%) | 7 | (10.9%) | 16 | (6.3%) | <0.001** | | | |
| Comorbidity | | | | | | | | | | |
| Osteoarthritis | 258 | (69.0%) | 41 | (64.1%) | 176 | (68.8%) | 0.730 | | | |
| Rheumatoid arthritis | 28 | (7.5%) | 10 | (15.6%) | 23 | (9.0%) | 0.104 | | | |
| Diabetes mellitus | 92 | (24.6%) | 12 | (18.8%) | 93 | (36.3%) | 0.001** | 0.345 | 0.005** | 0.011* |
| Hypertension | 185 | (49.5%) | 28 | (43.8%) | 185 | (72.3%) | <0.001** | 0.421 | <0.001** | <0.001** |
| Stroke | 59 | (15.8%) | 10 | (15.6%) | 67 | (26.2%) | 0.004** | 1.000 | 0.005** | 0.152 |
| Hyperthyroidism | 25 | (6.7%) | 7 | (10.9%) | 19 | (7.4%) | 0.483 | | | |
| Chronic liver disease | 65 | (17.4%) | 13 | (20.3%) | 63 | (24.6%) | 0.086 | | | |

By Kruskal Wallis test.

[§]Post-hoc analysis by Dunn-Bonferroni test.

* *p* value < 0.05

** *p* value < 0.01

S: self-discontinuation; D: discontinuation by physician suggestion; U: unintended discontinuation

bisphosphonates. However, in the present study we investigated osteoporosis medications with different mechanisms of action. We identified the leading causes of discontinuation of osteoporosis therapy, which were self-discontinuation (54.0%), physician's recommendation to discontinue the drug (9.2%), and unintended discontinuation (36.8%). We also found that among patients who self-discontinued therapy the reasons were as follows: they missed the outpatient appointment (46.3%); drug factors (24.5%); they thought their osteoporosis had improved (15.2%); economic factors (8%); and the occurrence of other diseases (5.9%). In addition, we also identified a group of patients who discontinued anti-osteoporosis medication

**Table 3. Predictors for self-discontinuation of osteoporosis medication.**

| | Univariate | | | Multivariate | | |
|---|---|---|---|---|---|---|
| | OR | 95% CI | *p* value | OR | 95% CI | *p* value |
| Age | 0.94 | (0.93–0.96) | <0.001** | 0.95 | (0.93–0.96) | <0.001** |
| Gender (Male vs Female) | 0.41 | (0.28–0.61) | <0.001** | 0.48 | (0.31–0.74) | <0.001** |
| BMI | 1.07 | (1.02–1.11) | 0.003** | | | |
| Smoking | 0.57 | (0.35–0.94) | 0.026* | | | |
| Alcohol consumption | 0.97 | (0.49–1.91) | 0.926 | | | |
| Fracture site | | | | | | |
| Hip | 0.42 | (0.28–0.63) | <0.001** | 0.66 | (0.43–1.03) | 0.065 |
| Spine | 0.72 | (0.52–1.00) | 0.045* | 1.02 | (0.70–1.50) | 0.912 |
| Radius | 0.77 | (0.32–1.85) | 0.563 | | | |
| Humerus | 0.74 | (0.27–2.07) | 0.572 | | | |
| Other | 0.80 | (0.52–1.23) | 0.307 | | | |
| Bone mineral density (g/cm$^2$) | | | | | | |
| Lumbar spine | 1.33 | (0.87–2.05) | 0.187 | | | |
| Right femoral neck | 9.93 | (4.90–20.11) | <0.001** | | | |
| Left femoral neck | 5.18 | (2.76–9.72) | <0.001** | | | |
| T-Score | | | | | | |
| Lumbar spine | 0.98 | (0.88–1.09) | 0.677 | | | |
| Right femoral neck | 0.94 | (0.83–1.07) | 0.352 | | | |
| Left femoral neck | 1.07 | (0.95–1.22) | 0.274 | | | |
| Calcium supplement | 0.67 | (0.49–0.91) | 0.014* | 0.67 | (0.48–0.95) | 0.023* |
| Vitamin D supplement | 0.79 | (0.55–1.14) | 0.209 | | | |
| Medications | | | | | | |
| Raloxifene | 1.00 | | | 1.00 | | |
| Alendronate | 0.63 | (0.41–0.98) | 0.039* | 0.82 | (0.51–1.33) | 0.424 |
| Denosumab | 0.62 | (0.42–0.92) | 0.018* | 0.88 | (0.57–1.35) | 0.560 |
| Zoledronic acid | 0.71 | (0.35–1.44) | 0.339 | 1.01 | (0.47–2.19) | 0.976 |
| Teriparatide | 0.17 | (0.07–0.43) | <0.001** | 0.19 | (0.08–0.49) | 0.001** |

Logistic regression.

*$p < 0.05$

**$p < 0.01$.

due to physicians' advice. To maintain adherence to anti-osteoporosis medication, health education should involve not only patients but physicians from different subspecialties. Moreover, we found that patients on raloxifene were at a higher risk of self-discontinuation. We speculate that raloxifene might have been prescribed for patients with mild osteoporosis, and therefore, would tend to have been more likely to discontinue their medication. However, patients who discontinue raloxifene might have increased risks of decline in BMD, suggesting case management should focus on patients on raloxifene [23]. Moreover, our study used telephone interviews to identify causes of discontinuation, but in-person interview was used in previous studies [11, 12]. Differences in study designs and anti-osteoporosis medications may lead to diverse study results. However, telephone interviews can identify patients' loss to follow-up because of death. Further studies should focus on discovering risks for mortality in patients discontinuing anti-osteoporosis medication. Furthermore, the universal coverage of the national health insurance system in Taiwan is unique and the geriatric population in Taiwan has exceeded 10%. Hence, our results may not be extrapolated to other parts of the world.

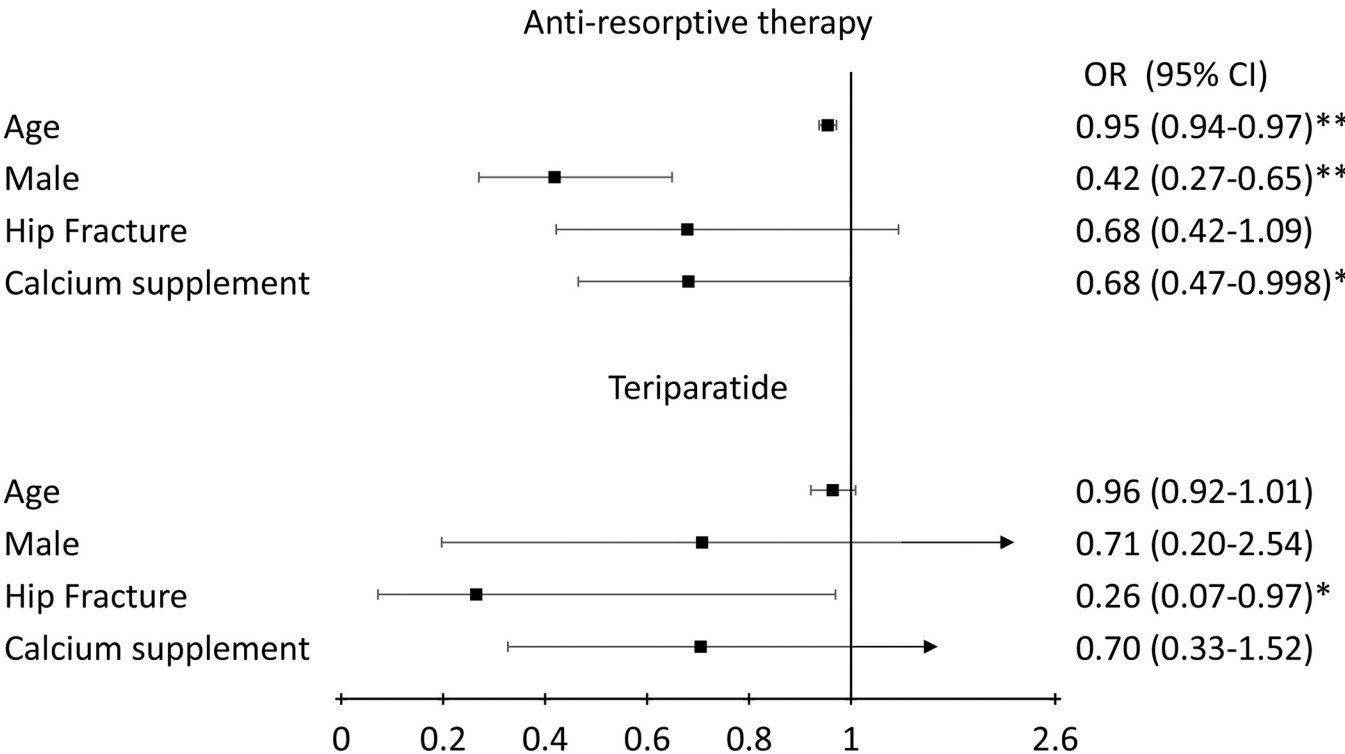

**Fig 2. Predictors for self-discontinuation of anti-osteoporosis medication by mechanisms of action.** * $p < 0.05$, ** $p < 0.01$.

However, we contend that our results might provide insight for case management of patients discontinuing anti-osteoporosis medications.

A previous study showed that the persistence rate of teriparatide at 12 months was 34.9%, and approximately one-third of the patients were not treated with any osteoporosis drugs after discontinuing teriparatide [24]. Our results indicate that patients with hip fractures were less likely to self-discontinue teriparatide compared with those without hip fractures. Additionally, participants with younger age, female gender, and not taking calcium supplements had a higher probability of stopping anti-resorptive therapy. The present study is the first to discover that subjects with younger age, female gender, and not taking calcium supplements tended to self-discontinue anti-osteoporosis drugs. We speculate that this group of patients might have a milder degree of osteoporosis. Because low and moderate adherence to osteoporosis medications appears to confer a higher risk of a subsequent fracture compared with high adherence patients [25], our results provide essential information for case management of osteoporosis in the future.

Although we discovered valuable factors associated with self-discontinuation of osteoporosis medication, several limitations should be considered. First, the causes of discontinuing osteoporosis medication were determined by telephone interviews, and therefore recall bias may have existed. Second, assessment of a geriatric population with impaired cognitive function or hearing loss could be problematic. Data could have been missed if calls were not answered. Frail elderly patients may have been under-represented in our study cohort. Meanwhile, the reasons why some physicians suggested stopping the osteoporosis medication were not explored. Third, the definition of discontinuation of each anti-osteoporotic medication may lead to selection bias. The study design prevented us from comparing discontinuation rates among various anti-osteoporosis medication. Lastly, because we only enrolled patients

who discontinued anti-osteoporosis drugs, it may not be possible to extrapolate our research results to patients who are still taking the medication. A prospective study is needed in the future to address this important issue.

## 5. Conclusion

This study identified that patients with younger age, female gender, non-hip fractures in teriparatide users, not taking calcium supplements, and anti-resorptive therapy were associated with self-discontinuation of osteoporosis medication. Some patients stopped taking the osteoporosis medication due to clinicians' recommendations. We believe that health education should involve physicians of all subspecialties to identify high-risk patients, with the goal of preventing subsequent fractures following discontinuation of osteoporosis medication.

## Acknowledgments

We would like to thank the Biostatistics Task Force of Taichung Veterans General Hospital for their assistance in performing the statistical analysis.

## Author Contributions

**Data curation:** Ya-Lian Deng, Chun-Hsi Shih.

**Methodology:** Chiann-Yi Hsu, Chih-Hui Chen, Chin-Feng Liu.

**Resources:** Shu-Hui Yang.

**Software:** Chiann-Yi Hsu.

**Supervision:** Shiang-Ferng Ou, Hsu-Tung Lee.

**Writing – original draft:** Ya-Lian Deng.

**Writing – review & editing:** Chun-Sheng Hsu, Yi-Ming Chen.

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
