## [Decision Letter · Decision Letter 0]

14 Jul 2022

PONE-D-21-34699Predictors for Self-Discontinuation of Anti-osteoporosis Medication: a Hospital-based Real-world StudyPLOS ONE

Dear Dr. Chen,

Thank you for submitting your manuscript to PLOS ONE. After careful consideration, we feel that it has merit but does not fully meet PLOS ONE’s publication criteria as it currently stands. Therefore, we invite you to submit a revised version of the manuscript that addresses the points raised during the review process.

We look forward to receiving your revised manuscript.

Kind regards,

Maria Maddalena Sirufo

Academic Editor

PLOS ONE

Journal Requirements:

2. Thank you for stating the following financial disclosure: "No" 

Additional Editor Comments (if provided):

Specify better why denosumab is the drug most subject to interruption, as in our experience it is the most tolerated and least subject to interruption.

specify the concept

Reviewers' comments:

Reviewer's Responses to Questions

**Comments to the Author**

1. Is the manuscript technically sound, and do the data support the conclusions?

Reviewer #1: Partly

Reviewer #2: Partly

2. Has the statistical analysis been performed appropriately and rigorously? 

Reviewer #1: Yes

Reviewer #2: Yes

3. Have the authors made all data underlying the findings in their manuscript fully available?

Reviewer #1: Yes

Reviewer #2: Yes

4. Is the manuscript presented in an intelligible fashion and written in standard English?

Reviewer #1: No

Reviewer #2: Yes

5. Review Comments to the Author

Reviewer #1: lines 8-10: "Comparisons among groups and risks of self-discontinuation were analyzed. Among 695 patients, 375 (54%) self-discontinued, 64 (9.2%) discontinued due to physicians’

suggestion, and 256 (36.8%) died."

The death may not be considered a cause of discontinued therapy. The authors may explain better the concept.

paragraph 2.2: the authors considered a cause of discontinuation of medication the death. This event can't be compared with the others two causes (self-discontinuation and discontinuation due to the physician’s recommendation) if the scope of this study is "to investigate risk factors and causes of poor

medication adherence in patients with osteoporosis, with a view to improving medication persistence."

Please expose better the concept.

Reviewer #2: Dear authors,

Thank you for completing this work for me to review.

Very interesting cohort study, PONE-D-21-34699, entitled “Predictors for Self-Discontinuation of Anti-osteoporosis Medication: a Hospital-based Real-world Study”. Only two questions were aroused while reading through your work.

First, what were the definitions of discontinuation of all the medication except for the denosumab and teriparatide mentioned in the method part? Do you think it is more reasonable to set up the definition of discontinuation according to the half-life of each of the selected medication? Or, please state your criteria accordingly.

Second, why the discontinuation rate of denosumab is the highest among all the medications, which is contrary to our clinical experiences, especially compared to those short half-life oral medications? Is it due to the definition of inclusion criteria or selection bias?

Thank you.

6. PLOS authors have the option to publish the peer review history of their article (what does this mean?). If published, this will include your full peer review and any attached files.

Reviewer #1: No

Reviewer #2: **Yes: **Yu-Jih Su, MD, PhD.

---

## [Author Response · Author response to Decision Letter 0]

6 Sep 2022

Dear reviewers and editor,

 We would like to express our sincere thanks to you and the reviewers for your thorough review of our manuscript and for the opportunity to submit a revised and improved version. We believe that by addressing the concerns, we have considerably improved our manuscript. We have provided point-by-point responses to the reviewers’ comments. Below is the author’s response to reviewer’s comments and suggestions.

Reviewer #1

Point 1: lines 8-10: "Comparisons among groups and risks of self-discontinuation were analyzed. Among 695 patients, 375 (54%) self-discontinued, 64 (9.2%) discontinued due to physicians’ suggestion, and 256 (36.8%) died." The death may not be considered a cause of discontinued therapy. The authors may explain better the concept.

Author’s response: Thank you for the comment. We agree with the reviewer’s opinion that the death may not be considered a cause of self-discontinuation of anti-osteoporosis therapy. Mortality should be considered as unintended discontinuation of treatment. Patients enrolled in the unintended discontinuation group mainly included patients with good adherence to anti-osteoporotic medication before death, and they stopped using anti-osteoporotic medication due to mortality. To avoid confusion, we have revised the aim of our study as “to investigate the causes of anti-osteoporosis medication discontinuation and to explore risk factors for self-discontinuation” (Please see page 5, lines 15-17). We also revised Table 2 and Figure 1 to re-categorize the study population. 

Point 2: paragraph 2.2: the authors considered a cause of discontinuation of medication the death. This event can't be compared with the others two causes (self-discontinuation and discontinuation due to the physician’s recommendation) if the scope of this study is "to investigate risk factors and causes of poor medication adherence in patients with osteoporosis, with a view to improving medication persistence." Please expose better the concept.

Author’s response: Thank you for the suggestion. We agree with the reviewer’s comment that the death can’t be compared with the others two causes if the scope of this study is "to investigate risk factors and causes of poor medication adherence in patients with osteoporosis. We believed that mortality contributes to medication discontinuation and should be classified as unintended discontinuation. As mentioned in the previous reply to reviewer’s comment, we have revised the aim of our study, and re-classify this group to “unintended discontinuation” (Please see page 7, lines 13-14, Table 2 and Figure 1). 

 

Reviewer #2

First, what were the definitions of discontinuation of all the medication except for the denosumab and teriparatide mentioned in the method part? Do you think it is more reasonable to set up the definition of discontinuation according to the half-life of each of the selected medication? Or, please state your criteria accordingly.

Author’s response: Thank you for bringing up this critical comment. We have added the definition of bisphosphonate or raloxifene discontinuation as ≥12 months without any anti-osteoporosis medication prescription claims after prior bisphosphonate or raloxifene treatment (please see page 6, line 26; page 7, lines 1-2). We agree with the reviewer’s opinion that it seems to be reasonable to set up the definition of discontinuation according to the half-life of each of the selected medication. However, the dosing frequency may last beyond the half-life of each medication in blood, i.e., biologic effect might be different from pharmacokinetics of medication. For example, the half-life of denosumab is less than 30 days but the dosing frequency is every 6 months. Therefore, we defined the duration of discontinuation of each anti-osteoporosis medication according to previous publications [1-5].

Reference

1. Silverman SL, Siris E, Kendler DL, Belazi D, Brown JP, Gold DT, et al. Persistence at 12 months with denosumab in postmenopausal women with osteoporosis: interim results from a prospective observational study. Osteoporos Int. 2015;26(1):361-72. Epub 2014/09/23. doi: 10.1007/s00198-014-2871-6. PubMed PMID: 25236877; PubMed Central PMCID: PMCPMC4286624.

2. Migliaccio S, Francomano D, Romagnoli E, Marocco C, Fornari R, Resmini G, et al. Persistence with denosumab therapy in women affected by osteoporosis with fragility fractures: a multicenter observational real practice study in Italy. J Endocrinol Invest. 2017;40(12):1321-6. doi: 10.1007/s40618-017-0701-3. PubMed PMID: 28589380.

3. Burge R, Sato M, Sugihara T. Real-world clinical and economic outcomes for daily teriparatide patients in Japan. Journal of bone and mineral metabolism. 2016;34(6):692-702. Epub 2016/11/01. doi: 10.1007/s00774-015-0720-0. PubMed PMID: 26661475.

4. Chen Q, Guo M, Ma X, Pu Y, Long Y, Xu Y. Adherence to Teriparatide Treatment and Risk of Fracture: A Systematic Review and Meta-Analysis. Horm Metab Res. 2019;51(12):785-91. doi: 10.1055/a-1062-9447. PubMed PMID: 31826274.

5. Adami G, Jaleel A, Curtis JR, Delzell E, Chen R, Yun H, et al. Temporal Trends and Factors Associated with Bisphosphonate Discontinuation and Restart. J Bone Miner Res. 2020;35(3):478-87. doi: 10.1002/jbmr.3915. PubMed PMID: 31714637; PubMed Central PMCID: PMCPMC7401723.

Second, why the discontinuation rate of denosumab is the highest among all the medications, which is contrary to our clinical experiences, especially compared to those short half-life oral medications? Is it due to the definition of inclusion criteria or selection bias?

Author’s response: Thank you for the comment. In our study, patients discontinuing denosumab accounts for the majority among 3 groups of different discontinuation causes (Table 2). It could be resulted both from the inclusion criteria and selection bias. We enrolled participants who discontinued anti-osteoporosis medication between June 2016 and June 2018. Because the denominators of the discontinuation rates were not calculated, the highest proportion that denosumab accounted for might be due to the largest market share of it. However, our retrospective study design prevented us from comparing discontinuation rates among various anti-osteoporosis medication. We discussed this point in the limitation section (please see page 17, lines 9-12).

---

## [Editor Report · Decision Letter 1]

9 Sep 2022

Predictors for Self-Discontinuation of Anti-osteoporosis Medication: a Hospital-based Real-world Study

PONE-D-21-34699R1

Dear Dr. Chen,

We’re pleased to inform you that your manuscript has been judged scientifically suitable for publication and will be formally accepted for publication once it meets all outstanding technical requirements.

Kind regards,

Maria Maddalena Sirufo

Academic Editor

PLOS ONE
---

## [Editor Report · Acceptance letter]

12 Sep 2022

PONE-D-21-34699R1 

Predictors for Self-Discontinuation of Anti-osteoporosis Medication: a Hospital-based Real-world Study 

Dear Dr. Chen:

I'm pleased to inform you that your manuscript has been deemed suitable for publication in PLOS ONE. Congratulations! Your manuscript is now with our production department. 

Kind regards, 

on behalf of

Dr. Maria Maddalena Sirufo 

Academic Editor

PLOS ONE